# Assessment of Ecosystem Service Supply, Demand, and Balance of Urban Green Spaces in a Typical Mountainous City: A Case Study on Chongqing, China

**DOI:** 10.3390/ijerph182011002

**Published:** 2021-10-19

**Authors:** Chang Luo, Xiangyi Li

**Affiliations:** 1Department of Landscape Architecture, College of Horticulture and Landscape Architecture, Southwest University, No. 2 Tiansheng Road Beibei, Chongqing 400715, China; llc555118@swu.edu.cn; 2Department of Landscape Architecture, School of Architecture and Urban Planning, Shenzhen University, Nanshan Avenue 3688, Shenzhen 518060, China

**Keywords:** ecosystem service, green spaces, supply–demand relationship, urban sustainability

## Abstract

Objective measurement of the supply–demand of ecosystem services (ESs) has received increasing attention from recent studies. It reflects the relationship between green spaces and human society. However, these studies rarely assess the mountainous cities. To fill this gap, this study takes a typical mountainous city as a research case to reveal the supply–demand relationship of ecosystem services, then development and management strategies are proposed for different districts according to their spatial differentiation characteristics. Results shows that: (1) there are differences of ESs supply between each district, and supply from Banan District is significantly higher than others. (2) The demands for ES also vary widely, which are higher in the core urban areas. (3) There are different degrees of imbalance between supply and demand in each district. We classified green spaces into four types based on their supply–demand characteristics, and optimization strategies are proposed. We found that most of the districts are lack of ES supply while there is a relatively high demand for ES in Chongqing, and the balance of supply and demand between different districts varies greatly. Our study indicates that targeted urban green spaces strategies for different districts must be considered to adequately optimize ES in mountainous cities.

## 1. Introduction

As important bridges linking natural environment and human societies, ecosystem services (ES) refers to the structures and functions for human well-being [1,2,3,4,5,6,7,8]. In many cities, urban greenspace (UGS) provides major ecosystem services for human societies [9,10,11]. Rapid urbanization has brought large-scale changes of urban greenspaces, which has led to the imbalances of supply and demand of ecosystem services in urban areas [12,13]. On the one hand, the imbalance between supply and demand is reflected by the lack of ES supply capacity: the encroachment of urban development and the loss of green spaces directly lead to the decline of ES supply [14,15,16,17]. According to the Millennium Ecosystem Assessment, more than 60% of the ecosystem services continue to degrade [6]. On the other hand, the imbalance is related to the uneven distribution of demand: with the increasing material wealth of urban residents, the demand for high-quality living environment is rising, especially in highly constructed areas [14,17,18]. With continually declining supply and rising demand, the contradiction between natural environment and human society has become more acute [3,19,20]. Focusing only on changes in ES supply, while ignoring changes in ES demand will lead to the imbalance of ES supply–demand, and thus causing a series of ecological degradation and social equity issues [3,19,20,21,22,23,24,25]. Thus, integrating ecosystem services into urban green space planning is crucial for future dwellers and decision-makers of the city [26,27,28,29]. For example, in China, urban green spaces development is emphasized to secure ecosystem services by national regulations such as the guideline of “Territorial Spatial Planning System” [30] and the guidelines for setting “Ecological Red Lines (ERL)” [31]. However, the attention to demand perspective of ecosystem services is still insufficient in these government documents and guidelines.

At present, there are practical problems remain in the process of urban green space planning: (1) Zoning is still the most common mode of urban planning, which has obvious advantages, like predictability of outcomes and ease of administration [29]. However, the lack of flexibility and the ability to adapt to changing social, ecological, and economic conditions set up the limitations of urban planning, which have led to increasing conflicts in urban areas [29]. (2) Urban green space is significantly affected by human activities, especially with rapid urbanization [15,32]. Although assessment of ecosystem services was integrated to planning regulations, the current assessment framework mainly focuses on the supply perspective of ecosystem services, while few consider the demand. The lack of a demand perspective may lead to a series of ecological degradation and social equity issues [33]. Therefore, it is necessary to map the supply and demand for ES, which can help to provide efficient land-use strategies and planning policies [34,35,36,37].

‘Mountainous cities’ refers to cities with a large proportion of hilly area or cities situated in isolated and narrow inner basins and plateaus surrounded or backed by mountains [38]. Nearly 1/10 of the world’s population live in mountainous settlements [39], and 1/3 of Chinese cities are defined as mountainous cities. Pressure from urbanization in mountainous cities is more intensive. For example, in mountainous cities, there are a lot of fragile and sensitive ecological areas, such as floodplains, water catchments, and steep hillsides, which were prone to ecological disasters [40]. However, mainstream studies have paid little attention to the ES supply–demand in mountainous cities.

To address these research gaps, this study proposed an analytical framework for measurements of ES supply and demand in mountainous cities. This study aims to develop an integrated approach to quantify and compare the supply and demand of multiple ESs: (1) Spatial quantification of the supply and demand of ecosystem services, (2) the degree of spatial match between supply and demand, and (3) optimization of policy-making for each districts. We apply the framework to the mountainous city of Chongqing, and we quantify the supply of seven ESs. Then, we collect the nighttime light data and estimate the demand for ESs. The matches and mismatches of the supply and demand of ESs are further identified by quadrant analysis. Finally, the special optimization strategies are discussed, and planning suggestions for enhancing the balance of supply and demand are presented.

## 2. Theoretical Basis and Conceptual Framework

### 2.1. Theoretical Basis and Definition of ES Supply and Demand

Ecosystem services refers to the goods and services of ecosystem structures and functions for human well-being [1,2,3,4,5,6,7,8]. ES research has experienced rapid development for decades, from focusing on ES supply to the emphasis on the balance and demand of ES [12,13]. However, many existing ES supply–demand studies focus on macro scale [12,41,42], ES supply–demand within cities, especially in mountainous cities, lags behind. Due to the different study scales, ecosystem supply and demand analysis is insufficient, and a challenge remains to accurately quantify ecosystem supply and demand [12,20].

According to previous studies, the definition of ES supply is relatively common, referring to the capacity of a particular area to provide ecosystem goods and services in a given time [3], while there are three main definitions of demand for ESs [23]. The demand for ESs can be defined as: (1) the sum of all actual use or consumption of ESs within a certain area over a certain period [3]; (2) desires and preferences for ESs as the amount of a service required or desired by society, including the demand for risk reduction (e.g., erosion control) [43]; and (3) the amount of a service required or desired by society, which should be further distinguished according to time and space [20]. The first emphasizes actual consumption relying on empirical data [17], while the second tends to identify the demand according to people’s willingness [21,41,44]. However, ecosystem service is the bridge between objective natural environment and subjective human society, thus, in this study, the third definition of the demand for ecosystem service has been adopted.

### 2.2. Determinants of Ecosysterm Services in Mountainous Cities

Mountainous cities refer to cities with a large proportion of hilly area or cities situated in isolated and narrow inner basins and plateaus surrounded or backed by mountains [45]. Populations in the mountainous cities of Europe and US are relatively small, because only a small portion of the population lives in the mountainous regions, mainly retirees and temporary residents [45]. In contrast, mountainous cities in China significantly differ from the leisure-oriented cities in the Europe and US. Rapid urbanization and rural migration have led to significant changes of urban greenspaces and natural environment, which has occurred the imbalances of ES supply and demand in urban areas [12]. According to previous studies, mountainous cities have complex urban environments that are different from the cities on flatlands, because of the interaction between intensive urban development and extremely restricting natural limits [35,38,39,45]. Meanwhile, according to the natural mountainous environment, the urban forms of mountainous cities naturally exhibit a polycentric structure, which is usually composed of several subcenters and urban clusters [45,46]. Thus, assessment of ES supply and demand of mountainous cities with the unique natural characteristics and the polycentric forms is crucial to sustainable greenspace management and decision-making for planning. To address these research gaps, this study developed a conceptual framework for measurements of ES supply and demand in mountainous cities.

### 2.3. Conceptual Framework

To address this research gap, a measuring system for ES supply–demand patterns was developed in mountainous cities based on existing studies. In particular, we proposed an operational method of measuring the demand of ES supply: i.e., we used the nighttime light data. This study aimed to propose an operational framework for mapping ES supply–demand patterns in mountainous cities. First, we assessed the ES supply based on multi-criteria: i.e., water supply for provision services; water regulation, water purification, carbon cycling, air purification, temperature regulation for regulating and supporting services, and recreation function for cultural services. Then, the nighttime light data were collected to evaluate the demand for ESs, and quadrant analysis was used to evaluate the matches between ESs supply and demand. Finally, we applied the ES supply and demand patterns to identify the imbalance of urban ecosystem. The research framework is shown in Figure 1. The results for mapping ES supply–demand patterns can provide operational suggestions for policy decision-making, especially in green space planning and management.

## 3. Materials and Methods

### 3.1. Study Area

We chose Chongqing city proper as the case. Chongqing is the one of the municipalities directly under the State Council in western China and a typical mountainous city on the upper reaches of the Yangtze River. The study area covers nine core urban districts of Chongqing, including Yuzhong, Jiangbei, Nanan, Shapingba, Dadukou, Jiulongpo, Yubei, Banan, and Beibei (Figure 2). Rapid urbanization in recent decades has caused a series of environmental issues in the core urban area of Chongqing—such as soil erosion, air pollution, flooding, and heat island effect [45,46].

### 3.2. Selections of Indicators and Data Sources

We extracted urban green spaces from remote sensing data and land use survey data of Chongqing in the years of 2015. The land use survey data from land and resources Bureau (1:10,000) were used to derive green space information to ensure the uniformity and accuracy of analysis. Air quality data are from the monitoring sites of local environmental protection department. Statistical data were obtained from the corresponding statistical yearbook [47]. Remote sensing data are mainly from SPOT data, and the city nighttime light data were mainly NPP/VIIRS data from Earth Observation Group. The land use/cover (farmland, woodland, grassland, wetland, water area, built-up area, unused land) were identified using the support vector machine supervised classification.

We selected the following seven indicators for mapping the supply for ESs according to the study of MEA [6]. They are selected from three dimensions of provision services, regulating and supporting services, and cultural services: (1) Provision services indicators include fresh water supply, which is a significant function of green spaces in a city with mountains and water [35,36,48]. (2) Regulating and supporting services include water regulation, water purification, carbon cycling, air purification, and temperature regulation. Water regulation services plays an important role in preventing urban flood, which is particularly severe in floodplains and low-lying areas of a mountainous city [14,35,45,49]. Water purification function reflects the regulation ability and stability of urban ecosystems [22]. Urban green space is an important component in global carbon cycling [50]. Air quality severely affects human health. In mountainous cities, the temperature inversion effect has impeded air flow and in turn intensified air pollution [51]. Temperature regulation is also included because the extreme heat has adverse effects on human health around the world [35], and the problem is even more serious due to the temperature inversion effect in mountainous cities [52]. (3) Cultural services are reflected by the recreation services of green spaces [10,53,54,55]. Nature-based recreation is an important function of ecosystems since it benefits people in many ways—such as increasing aesthetic experience [56], improving physical and mental health [57], and enhancing social cohesion [58]. In this study, we combined physical assessment method with the value assessment method to evaluate the value of ESs. The selected indicators, data source, and data processing are shown in Table 1.

### 3.3. Quantifying the Supply for ESs

#### 3.3.1. Provision Services

In this study, the data for provision services were directly from the local government statistics, and the value of water supply and food production were derived from the market value. The value of agricultural production was adopted from Chongqing Statistics Yearbook [47], while the value of fresh water could be calculated as
$$ESV_{w}=Q_{i}*C_{w}$$
where $ESV_{w}$ is the total value of fresh water supply service, $Q_{i}$ is the amount of water supply per year (m^3^), $C_{w}$ is the price of water resource according to the shadow price of water in China.

#### 3.3.2. Water Regulation

The water balance method was used to estimate the adopted to calculate the value of the stormwater regulation service. The formula is
$$ESV_{r}=\overset{n}{\underset{i=1}{\sum }}10A_{i}\left(P_{i}-E_{i}\right)*C_{r}$$
where $ESV_{r}$ is the value of water regulation, $C_{r}$ is the unit water storage cost in China, and i is the year, and $P_{i}$ is the annual precipitation (mm), $E_{i}$ is the annual evaporation of green spaces, $A_{i}$ is the green areas, and i is the year.

#### 3.3.3. Water Purification

It was demonstrated that green spaces have a positive impact on water purifying while water regulating. Therefore, the amount of water purification and its value can be calculated as
$$ESV_{p}=10A_{i}\left(P_{i}-E_{i}-R\right)*C_{p}$$
where $P_{i}$ is the annual precipitation (mm), $E_{i}$ is the annual evaporation of green spaces, *R* is the surface runoff, $A_{i}$ is the green space areas, $C_{p}$ is the unit water purification cost, and i is the year. According to the local sewage treatment price, the unit water purification cost of Chongqing is 2.78 yuan/m^3^.

#### 3.3.4. Air Purification

According to the ecological benefits of green spaces, the air purification service is including removing sulfur dioxide (SO_2_), Nitrogen oxide (NO_x_), and particulate materials (PM_10_), the value of air purification could be calculated as
$$ESV_{a}=\overset{n}{\underset{j=1}{\sum }}Q_{j}\times C_{j}\times A_{i}$$
where $ESV_{a}$ is the total value of air purification, $Q_{j}$ is the unit area air purification per year (kg), $C_{j}$ is the unit purification cost for different types of air pollutant, and j is the type of air pollutant, $A_{i}$ is the green areas, and i is the year.

#### 3.3.5. Temperature Regulation

Previous studies have shown that every 432.804 kJ of heat absorbed by the urban green space is equivalent to 20.64 kW·h of electricity consumed by air conditioning [55]. Therefore, the value of temperature regulation service was estimated by the market value method, and the formula is
$$ESV_{t}=\frac{Q_{t}}{432.804}\times 20.64\times F_{i}\times A_{i}$$
where $ESV_{t}$ is the value of climate regulation, $Q_{t}$ is the heat absorption by evaporation, $F_{i}$ is the electricity fee in Chongqing, and $A_{i}$ is the area of green space.

#### 3.3.6. Carbon Cycling

Through photosynthesis, plants have a positive effect on CO_2_ sequestration and O_2_ release. At the city scale, the CO_2_ sequestration and oxygen release of green spaces can be calculated by the net primary productivity (NPP) reflecting the carbon ‘metabolism’ of urban vegetation, and the calculation formula is
$$ESV_{c}=G_{c}\times C_{c}+G_{O}\times C_{O}$$
$$G_{c}=1.63\times R_{c}\times A_{i}\times B_{i}$$
$$G_{O}=1.19\times A_{i}\times B_{i}$$
where $ESV_{c}$ is the total value of carbon cycling, $G_{c}$ and $G_{O}$ are the amount of carbon sequestration and oxygen release; $C_{c}$ and $C_{O}$ are the carbon tax price and unit oxygen price. $R_{c}$ is the carbon ratio in CO_2_, $B_{i}$ is the average NPP in a year, $A_{i}$ is the green space areas, and i is the year.

#### 3.3.7. Recreation Service

In this study, due to the limitation of data resource, the cultural service is main reflected by the recreation value of green spaces. The assessment of recreation services is based on the data for visitor numbers and the ticket price of parks, and the number of visitors was mainly from Chongqing Municipal Bureau of Landscape Architecture, and the formula is
$$ESV_{i}=\overset{n}{\underset{i=1}{\sum }}Q_{k}\times C_{k}$$
where $ESV_{i}$ is the total value, $Q_{k}$ is the visitor number of parks, $C_{k}$ is the ticket price of parks, k is the type of park, and i is the year.

### 3.4. Quantifying the Demand for ESs

Due to the different definitions of demand for ESs, the assessment methods vary across studies. Some of the previous studies have assessed the demand for ESs from the aspect of preferences and desires using the method of public participation [56,59], which allows direct feedback from the human society, while the others mainly use socioeconomic indicators to qualify the ESs demand [42,60]. In this study, the nighttime light data were used as a proxy for the demand of ESs. Nighttime light data have been widely used to study economic activity, land-use change, and settlement dynamics. Compared with specific socioeconomic indicators (e.g., population density, economic activity) [61,62,63], nighttime light data can reflect the demand for ESs efficiently at the spatial scales [64]. This study used freely accessible NPP/VIIRS nighttime light data from January to December 2015 and the annual average nighttime light data of each district.

### 3.5. Analysis of the Relationship between Supply and Demand of ESs

In this study, we used quadrant analysis to evaluate the matches between ESs supply and demand [42,65]. Taking the district-level administrative unit as the basic unit, we applied the quadrant analysis to identify the classification of supply–demand relationships, and optimization strategies for each district were applied. First, Z-scores standardization was used to determine the matching pattern of supply–demand ecosystem services. Then, the standardized demand for ESs is plotted on the *x*-axis and the standardized supply is plotted on the *y*-axis, which generates four quadrants: quadrant I represents high supply-high demand (H–H) districts, quadrant II represents high supply-low demand (H–L) districts, quadrant III represents low supply-low demand (L–L), and quadrant IV represents low supply-high demand (L–H) districts. The formula is
$$X=\frac{x_{i}-\bar{x}}{s}$$
$$\bar{x}=\frac{1}{n}\overset{n}{\underset{i=1}{\sum }}x_{i}$$
$$s=\sqrt{\frac{1}{n}\overset{n}{\underset{i=1}{\sum }}{\left(x_{i}-\bar{x}\right)}^{2}}$$
where X is the standardized supply or demand for ESs, $x_{i}$ is the supply or demand of each evaluation unit, $\bar{x}$ is the average of all districts, *s* is the standard deviation, and n is the total number of districts. The spatial distribution of standardized supply or demand for ESs were overlaid in ArcGIS 10.3 to analyze the matching pattern.

## 4. Results

### 4.1. Special Pattern of ESs Supply

The total supply of ecosystem service value in the study area is about 54.32 billion yuan (Table 2). From Figure 3, districts with a high supply of ecosystem services in 2015 were mainly distributed away from the city center, while there was a low supply in the central districts of the city, especially in the Yuzhong district, which shows the lowest supply of ESs. Low supply districts were also distributed in the Jiangbei, Nanan, and Dadukou, due to the dense population and high proportion of highly constructed areas. In contrast, the districts of Banan, Yubei, and Beibei have a relatively large scale of green spaces, the higher supply of ecosystem services was mainly distributed in these areas.

Looking at the spatial distribution of different ESs types (Figure 4), Banan District and Beibei District have higher values of all types of services, while Shapingba District shows a higher value of cultural services. Due to its large scale of green spaces, a high supply of regulating and supporting services was distributed in Yubei district. In general, districts with high supply of ESs were scattered in the rural areas located in the south and north of the study area, while the districts with low supply of ESs were concentrated mostly at the center of the study area (Figure 5).

### 4.2. Special Pattern of ESs Demand

This study used nighttime light intensity to reflect the demand for ESs at the district scale (Figure 6). The demand for ecosystem services was mainly located at the center of the study area, corresponding with the city of Chongqing, especially in the Yuzhong district. There was also an intense demand for ESs in the west of Chongqing—including Shapingba, Jiangbei, Nanan, Dadukou, and Jiulongpo districts—where urban expansion was high. Meanwhile, the low demand districts were distributed at the suburb of the core city—including the districts of Banan, Yubei, and Beibei—due to their relatively low population density.

### 4.3. Spatial Matches between the Supply and Demand

Referring to the method of previous studies [14,17], the Z-Scores-standardization was used to standardize the ES supply and demand, and standardized supply and demand of ecosystem services was generated into four quadrants to determine the matching patterns. Based on the quadrant diagram of the supply–demand of ecosystem services, the matching supply–demand patterns of ecosystem services were divided into four types: High supply–high demand (H–H), high supply–low demand (H–L), low supply–low demand (L–L), and low supply–high demand (L–H). As seen in Figure 7, most districts of Chongqing city belonged to the H–L, L–L, and L–H patterns, while the districts of the H–H pattern are vacant.

According to the supply–demand patterns of ESs (Figure 8), the districts with the H–L pattern were mainly far from the city center in the north and southeast of the city, where the population and proportion of constructed area was low compared to other districts. In these districts, there are relatively large scale of greenspaces which are the main source of ecosystem services. The districts of Yubei, Beibei, and Banan are rich in forest resources, and their aggregate volume accounts for more than 80 percent of the city’s total. As the forest plays a significant role in regulating and supporting services, these districts are characterized by high supply of ESs.

The L–H pattern was mainly concentrated in central districts—including the Shapingba, Jiangbei, Nanan, and Yuzhong districts. These districts had a high population density, a high level of urbanization and a high consumption of ecosystem services. Developed land was one of the major land use types in these districts, where human activities generated great demand for ESs. Additionally, the lack of ecological land in these districts, such as forests and grassland, caused the supply capacity of ecosystem services was weak [34,56,66,67,68]. The imbalance between the supply and demand of ecosystem services was significant.

The districts of the L–L pattern were mainly located in the southwest part of the study area. In these districts, the population density was relatively low compared with the H–L districts, due to low economic development, and the demand for ecosystem services was relatively low. With industrial upgrading of the city, residents were gradually moving away from the old districts to the new developed districts, which caused the lower population concentration and slower economic growth in these districts. Meanwhile, most land of these districts was developed decades ago. Due to unreasonable development, such as water and soil pollution caused by steel industry [45], the supply of ecosystem services was also low.

The district with the H–H pattern was vacant in Chongqing city, reflecting the significant imbalance in supply and demand of ecosystem services. The social and economic development in the H–H pattern is at moderately high speed, while there is a stable and healthy ecosystem and high vegetation coverage [42,65]. At present, only Shapingba district was closest to this pattern.

## 5. Discussion

### 5.1. Spacial Imbalance and Optimization Strategies

According to the supply–demand patterns of ecosystem services, the districts of Chongqing city were divided into four types of relationship: H–H (high supply-high demand), H–L (high supply-low demand), L–H (low supply-high demand), and L–L (low supply-low demand), shown in Table 3. In general, most districts in our study area have different degrees of imbalance in supply and demand of ESs, which was consistent with that of the land use pattern of the study area [38,46,69]. Mismatches between the supply and demand of ES are common phenomena that demonstrate the lack of ecological resilience in modern cities [34], and these issues are even more serious for mountainous cities with the pressure of a sensitive environment [46].

In the districts with H–L pattern, the high proportion of greenspaces and decreased disturbance by human activities resulted in the high supply and low demand of ESs. For these districts, it is necessary to maintain the existing green coverage and natural ecosystems, restore vegetation in mountainous areas, set strict limits on land development, and prevent the disruption of urbanization.

Among the four types of patterns, the number of districts belonging to the L–H pattern was the largest, which reflects the imbalance between natural environment and human society. In this study, the districts of Yuzhong, Nanan, Shapingba, and Jiangbei located in the center area are old districts developed decades ago, which have a higher proportion of constructed land and population density. Due to the decades of urbanization, the supply of ecosystem services in these districts is lower than that in other districts. On the one hand, the imbalance was caused by the lack of supply capacity of ESs due to the encroachment of urban development and the fragmentation of green spaces. On the other hand, due to the over-concentration of urban population, the demand for ecosystem services was concentrated in the core districts. The acceleration of urbanization will aggravate the spatial imbalance between ecosystem services supply and human demand for natural environment [14,32,35]. For these districts, it is crucial to gradually increase the coverage and connectivity of green spaces through urban renewal, improve the ESs supply ability of existing urban green spaces, strictly control the expansion of construction land, and improve the efficiency of land use.

Jiulongpo and Dadukou belong to the L–L pattern, which were the early developed industrial districts. As well as the L–H pattern, these districts have a high proportion of constructed land and poor natural environment. However, with the decline of industrial enterprises, population density gradually decreased, which led to the low demand for ecosystem services in these districts. In the districts with L–L pattern, the need to restore natural ecosystems, remediate the problem of industrial pollution, and improve the supply of ecosystem services should be prioritized. At the same time, the potential of eco-friendly development should be exploited, and the optimization and improvement of industrial infrastructure should be considered to increase the urban attraction in these districts.

According to the results, the district with the H–H pattern was vacant in Chongqing city, which is a relatively balanced pattern in terms of supply and demand of ecosystem services. The lack of district with the H–H pattern demonstrated the significant imbalance in supply and demand of ecosystem services in the study area. Therefore, optimization and strategies should be enhanced to increase the supply capacity of ecosystem services to increase the proportion of districts with the H–H pattern in the city.

### 5.2. Research Limitations

In this study, statistics and nighttime light data were employed to reflect ecosystem service demand, which represented a spatial pattern of consumption and needs of human society in a city. The matching analysis of supply and demand represented a relative imbalance in spatial distribution, which provided an efficient analysis method for policy making and actual regional development, especially under the rapid urbanization of China.

However, there are also limitations in this study, for instance, the nighttime light data were adapted to quantify the demand for ecosystem services, which objectively reflect the special pattern of demand at the district level; however, the results were limited by the geographical data. The supply–demand of ecosystem services could be affected by various factors, including physical geographical factors or socioeconomic factors. It is necessary to use questionnaire data and field observation data in different districts to improve the analysis accuracy. Meanwhile, we only assessed the ecosystem services supply–demand patterns of 2015, mostly due to lack of available land survey data at the scale of our study. Thus, we should also focus on changes of supply–demand patterns and ecosystem service flows in future studies.

## 6. Conclusions

In this study, we proposed a multi-criteria approach to quantify the imbalance and spatial patterns of ES supply–demand. Particularly, we introduced the nighttime light data to identify the demand of ecosystem services. The studied results could contribute to sustainable greenspace management and decision-making for future land use planning. The results show that there are different degrees of imbalance in supply and demand of ecosystem services in all districts, especially in urban core areas. Balancing supply and demand of ecosystem services is crucial to sustainable urban greenspaces management and urban ecosystem protection. This study extends the work of previous studies and provides a practical approach for analyzing the supply–demand patterns of ecosystem services in the urban area. The study of local ecosystem services contributes to better understanding of ecological processes and improves the rationality of decision-making, leading to more efficient environmental management, sustainable urban development, and better protection of ecosystem functions.

## Figures and Tables

**Figure 1 ijerph-18-11002-f001:**
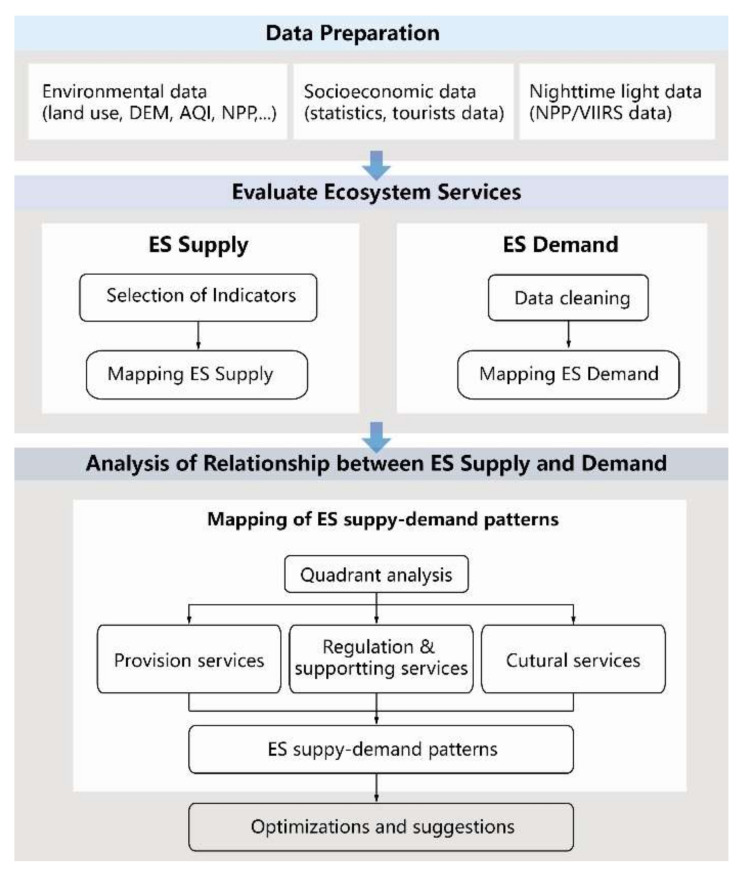
Conceptual framework.

**Figure 2 ijerph-18-11002-f002:**
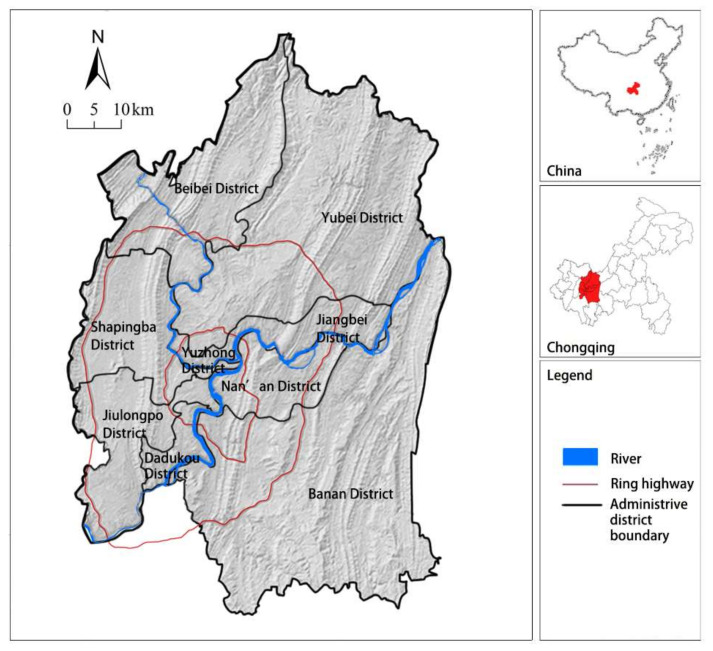
Study area and location.

**Figure 3 ijerph-18-11002-f003:**
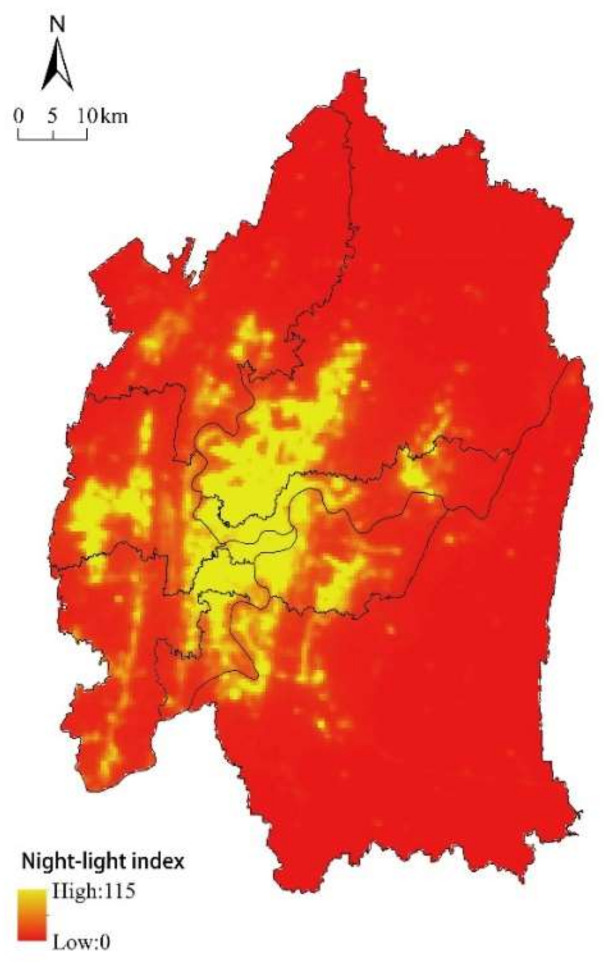
Nighttime light intensity in 2015.

**Figure 4 ijerph-18-11002-f004:**
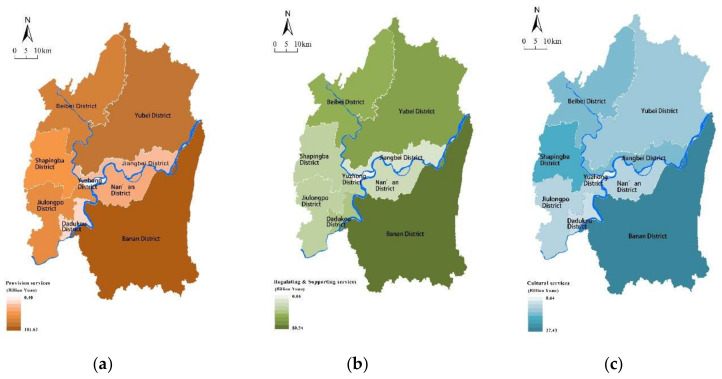
(**a**) Spatial distribution of provision services. (**b**) Spatial distribution of regulating and supporting services. (**c**) Scheme 5. Spatial distribution of ESs supply.

**Figure 5 ijerph-18-11002-f005:**
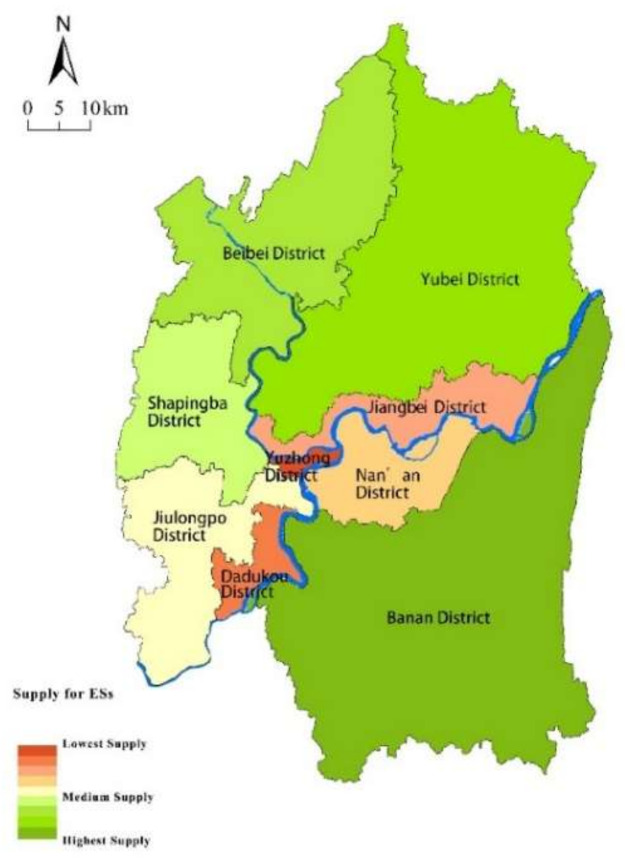
Spatial distribution of ESs supply.

**Figure 6 ijerph-18-11002-f006:**
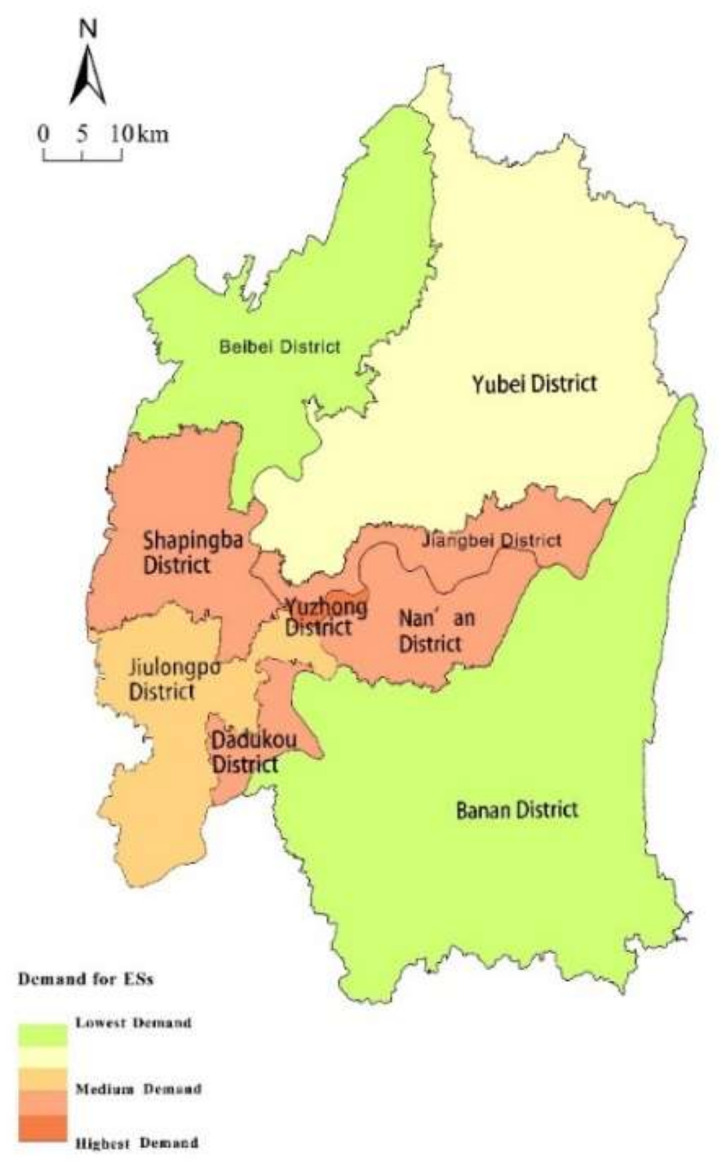
Spatial distribution of ESs demand.

**Figure 7 ijerph-18-11002-f007:**
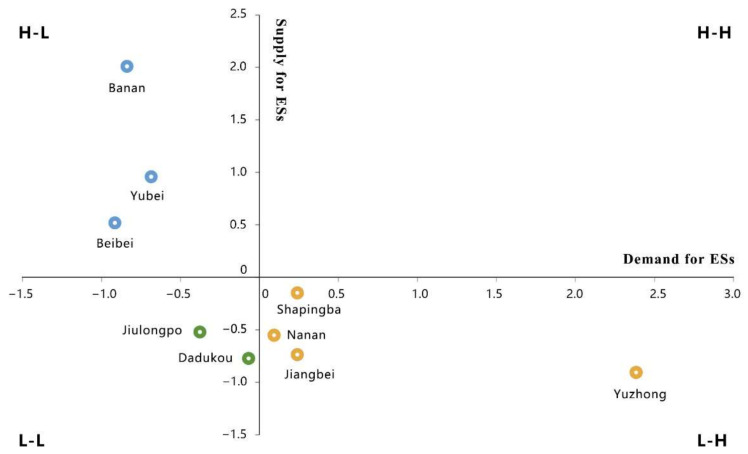
Supply–demand types of ESs.

**Figure 8 ijerph-18-11002-f008:**
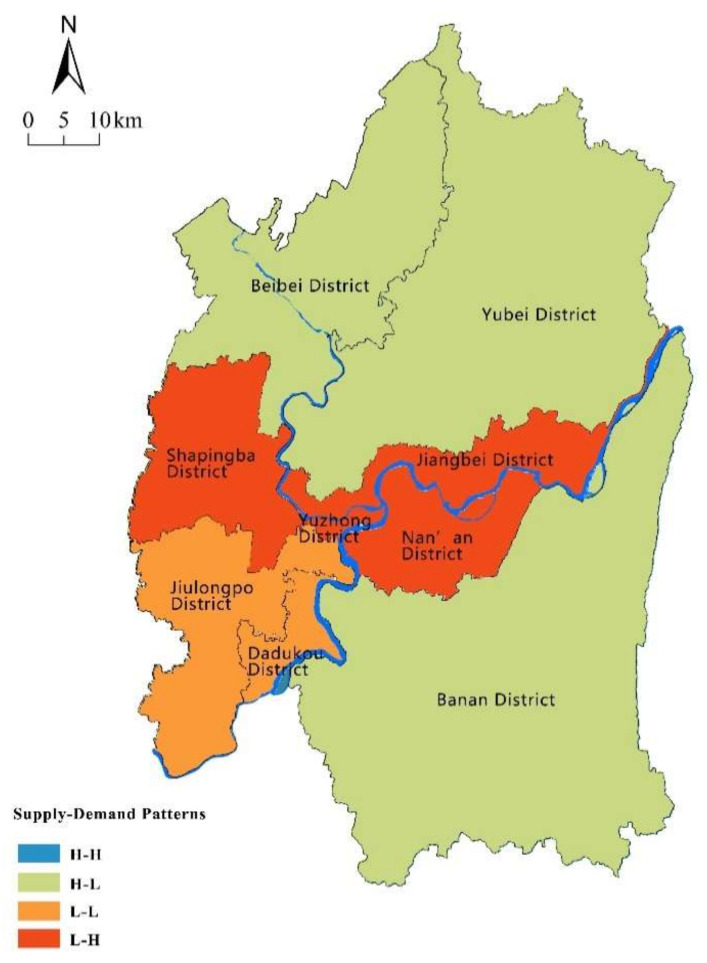
Distribution of supply–demand patterns of ESs.

**Table 1 ijerph-18-11002-t001:** Indicators and data source.

	Categories	Variables	Meaning	Data sources
Supply for Ecosystem Services	Provision services	Freshwater	Water supply (m^3•a−1^)	Chongqing Water Resources Bulletin (2015) (http://slj.cq.gov.cn/ accessed on 12 October 2018)
	Regulating and supporting services	Water regulation	Rainwater runoff consumption (m^3•a−1^)	Chongqing Water Resources Bulletin (2015) (http://slj.cq.gov.cn/ accessed on 12 October 2018)
		Water purification	Volume of purified water (m^3•a−1^)	
		Carbon cycling	Amount of carbon fixation and oxygen release (kg^•a−1^)	Net Primary Production (NPP) data of Chongqing
		Air purification	Purification amount of sulfur dioxide (SO_2_), nitrogen oxides (NO_x_), and PM_10_ (kg^•a−1^)	AQI data from 17 air monitoring stations in 2015
		Temperature regulation	Annual heat absorption of evaporation (kj^•a−1^)	Temperature data from Weather China Website (http://www.weather.com.cn/ accessed on 3 October 2018)
	Cultural services	Recreation service	The number of visitors to the park and the cost per person	Tourists Data from Chongqing Bureau of Parks
Demand for Ecosystem Services	Average demand for ESs	Nighttime lights	Nighttime lights (NTL) satellite imagery	NPP/VIIRS data from Earth Observation Group (https://payneinstitute.mines.edu/eog/ accessed on 8 November 2020)

**Table 2 ijerph-18-11002-t002:** Assessment results of supply for ecosystem services.

District	Provision Services	Regulating and Supporting Services	Culture Services	Total (Billion Yuan)	Percentage (%)
Yuzhong District	0.04	0.00	0.06	0.11	0.20%
Jiangbei District	0.53	0.34	0.36	1.23	2.27%
Nan’an District	1.29	0.69	0.52	2.49	4.58%
Yubei District	6.81	5.36	0.35	12.52	23.05%
Shapingba District	1.46	0.94	2.74	5.15	9.48%
Beipei District	3.69	3.75	2.08	9.52	17.52%
Jiulongpo District	1.84	0.66	0.26	2.75	5.06%
Dadukou District	0.44	0.42	0.21	1.07	1.96%
Banan District	10.16	8.93	0.39	19.48	35.86%
Total	0.00	0.00	0.00	54.32	100.00%

**Table 3 ijerph-18-11002-t003:** Classification of different districts based on supply–demand relationships.

Ecological Spaces Types	Supply–Demand Relationship of Ecosystem Services	Number	Districts
Ecosystem Coordination	High Supply–High Demand	0	-
Ecosystem Conservation	High Supply–Low Demand	3	Banan District, Beipei District, Yubei District
Ecosystem Restoration	Low Supply–High Demand	4	Nan’an District, Jiangbie District, Shapingba District, Yuzhong District
Ecosystem Reconstruction	Low Supply–Low Demand	2	Jiulongpo District, Dadukou District

## Data Availability

The data presented in this study are available on request from the corresponding author. The data are not publicly available due to part of them are being used in other studies that have not yet been publicly published.

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
