# Peer review of "Assessment of Ecosystem Service Supply, Demand, and Balance of Urban Green Spaces in a Typical Mountainous City: A Case Study on Chongqing, China"

_ijerph, 2021, doi:10.3390/ijerph182011002_

Round 1

Reviewer 1 Report

  1. The novelty should be mention in the Abstract
  2. Methodology should be improved and method used should be mentioned accurately with reason of that.
  3. In Table 1, why Data resourses is belong to 2015? Had not new data?
  4. Importance of Indicators should be mentioned before the section 2.3. Also, you can references to some of works belong to Armin Razmjoo in this regard.
  5. Where is the reference of the Fig 3. You have mention some data that needs to reference for this Fig.
  6. In 3.3 section, you have mentioned that you have used of Quadrant analysis. why? Do you think this is the best?
  7.  Before the conclusion, you should mention some suggestions that can be practical and positive effect of this work on environment should be mentioned.
  8. References between 2019- 2021 related to this work should be more mentioned especially in the Introduction.
  9.  All data presented should be carfullty checked to prevent possible mistakes.

Author Response

1.The novelty should be mention in the Abstract.

Studies rarely assess the supply-demand of ecosystem services of mountainous cities, and this paper try to focus on this kind of city, which has been added in the abstract, and added in the introduction part, from line 66 to 76.

2.Methodology should be improved and method used should be mentioned accurately with reason of that.

The reasons why these indicators have been selected, and the references have been explained in the second part, from line 109 to 125.

3.In Table 1, why Data resources is belong to 2015? Had not new data?

The latest Data opened to the public in China is belong to 2015.

4.Importance of Indicators should be mentioned before the section 2.3. Also, you can references to some of works belong to Armin Razmjoo in this regard.

We have added that from line 109 to 125,

5.Where is the reference of the Fig 3. You have mention some data that needs to reference for this Fig.

We have added assessment data in Table 2 that reference for Fig 3,

6.In 3.3 section, you have mentioned that you have used of Quadrant analysis. why? Do you think this is the best?

We add references in line 279-282 to explain why use quadrant analysis. “Referring to the method of previous studies[14, 17], the Z-Scores-Standardization was used to standardize the ES supply and demand, and standardized supply and demand of ecosystem services was generated into four quadrants to determine the matching patterns”

7.Before the conclusion, you should mention some suggestions that can be practical and positive effect of this work on environment should be mentioned.

In section 4, we discussed the positive effect of this work

8.References between 2019- 2021 related to this work should be more mentioned especially in the Introduction.

We add references between 2019- 2021 in the Introduction and discussion part.

9.All data presented should be carefully checked to prevent possible mistakes.

Thank you so much, we have checked them again.

Reviewer 2 Report

The article is very well structured and addresses a fundamental issue related to the measurement of the supply-demand of ecosystem services.

This subject has great global attention today, especially in times of climate change and its critical consequences in public life of cities. This type of studies are essential for cities and regions,  proposing novel approaches for mitigate its impacts. However, the article does not present a theoretical framework that addresses the main evidence of this type of studies worldwide. It is only limited of making a brief approach to the case of Chongqing in China, leaving out a lot of international evidence about how to evaluate and measure the supply-demand of ecosystem services in other contexts.

This is not a minor problem, since both the results and their discussion are not presented as a novelty in ​​environmental studies. Therefore, their potential contribution to the international state of the art is questionable. This is an International Journal. Therefore, the potential contribution of this article must be interest for readers worldwide.

My best suggestion to solve this problem is that the article should include a new section referred to the International State of the Art, before entering to Materials and Methods section. Likewise, authors should discuss the results in section  4) in relation with the international evidence from State of the Art analysis, describing potential of results, its limitations and contributions to the international state of the art.

Finally, the conclusions should describe limitations of this study and future research on these topic, as well as its real contribution to other international contexts and not only China.

Author Response

The article is very well structured and addresses a fundamental issue related to the measurement of the supply-demand of ecosystem services.

This subject has great global attention today, especially in times of climate change and its critical consequences in public life of cities. This type of studies are essential for cities and regions, proposing novel approaches for mitigate its impacts. However, the article does not present a theoretical framework that addresses the main evidence of this type of studies worldwide. It is only limited of making a brief approach to the case of Chongqing in China, leaving out a lot of international evidence about how to evaluate and measure the supply-demand of ecosystem services in other contexts.

We made a theory review and add more references in the introduction section to addresses the significance, practical issues and research gaps of this type of studies. We take Chongqing as a study case because of its typical characteristics of mountainous cities.

This is not a minor problem, since both the results and their discussion are not presented as a novelty in ​​environmental studies. Therefore, their potential contribution to the international state of the art is questionable. This is an International Journal. Therefore, the potential contribution of this article must be interest for readers worldwide.

Objective measurement of the supply-demand of ecosystem services (ESs) has received increasing attention from recent studies. However, these studies rarely assess the mountainous cities. Nearly one-tenth of the world’s population live in mountainous settlements, so to fill this gap, this study takes a typical mountainous city as a research case to reveal the supply-demand relationship of ecosystem services, then development and management strategies are proposed for different districts according to their spatial differentiation characteristics.

My best suggestion to solve this problem is that the article should include a new section referred to the International State of the Art, before entering to Materials and Methods section. Likewise, authors should discuss the results in section 4) in relation with the international evidence from State of the Art analysis, describing potential of results, its limitations and contributions to the international state of the art.

We revised the discussion section. We discussed the results in relation with other studies in section 4.1 and described our research limitation and future study in section 4.2.

Finally, the conclusions should describe limitations of this study and future research on this topic, as well as its real contribution to other international contexts and not only China.

We add limitations of this study and future research in section 4.

Round 2

Reviewer 2 Report

the manuscript has been imporved. However, it has been minimally improved regarding the discussion of the theoretical framework. The authors only incorporated a paragraph in the introduction about the concept of "Mountainous cities", without doing an analysis of opposition between authors that allows us to understand more evidence on the subject in question internationally. I believe that the article should add an additional section to strengthen the state of the art analysis.

Author Response

The manuscript has been improved. However, it has been minimally improved regarding the discussion of the theoretical framework.

Thank you so much for the suggestions. According to your suggestion, we add the new section before the materials and methods section to improve theoretical framework.

The authors only incorporated a paragraph in the introduction about the concept of "Mountainous cities", without doing an analysis of opposition between authors that allows us to understand more evidence on the subject in question internationally.

In section 2, we added theoretical reviews on ecosystem services and mountainous cities to discuss the research gaps and emphasis the insights of this study.

I believe that the article should add an additional section to strengthen the state of the art analysis.

Section 2 theoretical basis and conceptual framework has been added before methods section to discuss the research gaps and strengthen the state of the art analysis.
